# Comparative Molecular Mechanisms of Biosynthesis of Naringenin and Related Chalcones in Actinobacteria and Plants: Relevance for the Obtention of Potent Bioactive Metabolites

**DOI:** 10.3390/antibiotics11010082

**Published:** 2022-01-10

**Authors:** Juan F. Martín, Paloma Liras

**Affiliations:** Department of Molecular Biology, University of León, 24071 León, Spain; paloma.liras@unileon.es

**Keywords:** naringenin, chalcone synthases, flavonoids, actinobacteria, *Streptomyces clavuligerus*, aromatic acid starter units, ammonia lyase

## Abstract

Naringenin and its glycosylated derivative naringin are flavonoids that are synthesized by the phenylpropanoid pathway in plants. We found that naringenin is also formed by the actinobacterium *Streptomyces clavuligerus*, a well-known microorganism used to industrially produce clavulanic acid. The production of naringenin in *S. clavuligerus* involves a chalcone synthase that uses *p*-coumaric as a starter unit and a P**_450_** monoxygenase, encoded by two adjacent genes (*ncs-ncyP*). The *p*-coumaric acid starter unit is formed by a tyrosine ammonia lyase encoded by an unlinked, *tal*, gene. Deletion and complementation studies demonstrate that these three genes are required for biosynthesis of naringenin in *S. clavuligerus*. Other actinobacteria chalcone synthases use caffeic acid, ferulic acid, sinapic acid or benzoic acid as starter units in the formation of different antibiotics and antitumor agents. The biosynthesis of naringenin is restricted to a few *Streptomycess* species and the encoding gene cluster is present also in some *Saccharotrix* and *Kitasatospora* species. Phylogenetic comparison of *S. clavuligerus* naringenin chalcone synthase with homologous proteins of other actinobacteria reveal that this protein is closely related to chalcone synthases that use malonyl-CoA as a starter unit for the formation of red-brown pigment. The function of the core enzymes in the pathway, such as the chalcone synthase and the tyrosine ammonia lyase, is conserved in plants and actinobacteria. However, *S. clavuligerus* use a P**_450_** monooxygenase proposed to complete the cyclization step of the naringenin chalcone, whereas this reaction in plants is performed by a chalcone isomerase. Comparison of the plant and *S. clavuligerus* chalcone synthases indicates that they have not been transmitted between these organisms by a recent horizontal gene transfer phenomenon. We provide a comprehensive view of the molecular genetics and biochemistry of chalcone synthases and their impact on the development of antibacterial and antitumor compounds. These advances allow new bioactive compounds to be obtained using combinatorial strategies. In addition, processes of heterologous expression and bioconversion for the production of naringenin and naringenin-derived compounds in yeasts are described.

## 1. Introduction: Flavonoids and Chalcones, Naringenin and Related Compounds in Nature

Numerous plant, fungal and bacterial metabolites are formed through the phenylpropanoid pathway by type III polyketide synthases. These compounds include flavonoids, lignin and coumarins [1]. The flavonoids constitute a large group of biological products which are responsible for the colours of flowers and aromas of plants and protect them against UV irradiation and microbial pathogen infections [2,3]. Thousands of flavonoid compounds are known and many are important in human and animal nutrition since they are abundant in vegetables, fruits and nuts; besides, some of them are utilized in cosmetics [4]. Others show pharmacological activities [5], e.g., quercetin and kaempferol that have, respectively, antioxidant and antitumor activities [6].

One of these flavonoids is naringenin, a precursor of some strongly bitter glycosylated derivatives undesirable in citrus fruits, particularly in grape fruits [7]. Naringenin is an antioxidant that scavenges oxygen radicals, has anti-inflammatory properties [8,9,10] and is an inhibitor of two-pore channels in human cells [11]. Importantly, naringenin blocks the neo-angiogenesis, a process required for solid tumor progression [11] and is useful in the treatment of several viral infections [12].

Different plants produce naringenin and its biosynthesis was studied initially in parsley and in *Arabidopsis thaliana* [13,14,15]. The carbon backbone of flavonoids is formed by reiterated condensation of coenzyme A (CoA)-activated precursor units by chalcone synthases (CHSs). These enzymes are type III polyketide synthases that consist in a single polypeptide chain and are simpler than the type I and type II PKSs [16] as confirmed by recent crystallization studies [17]. An array of plant secondary metabolites is synthesized by different members of the chalcone synthases family [18]. Some chalcone synthases use distinct phenylpropanoid starter units (typically *p*-coumaric acid, caffeic acid and ferulic acid) as activated CoA derivatives (Figure 1) and elongate them by incorporation of malonyl-CoA units [19]. In contrast to type I PKSs, the chalcone synthases lack the phosphopantetheinyl chain [20], that in type I PKSs is attached to the acyl-carrier protein (ACP domain); instead, these chalcone synthases use directly CoA-activated units without transferring the acyl group to a phosphopantetheine carrier.

Recently, we found that naringenin is also synthesized by the actinobacterium *Streptomyces clavuligerus* [21], a well-known microorganism used in the industrial production of clavulanic acid, that also synthesizes the β-lactam antibiotic cephamycin C and several other bioactive compounds [22]. *S. clavuligerus* produces naringenin and its glycosylated derivative, monoglucosylnaringin [21,23]. The naringenin CHSs in plants and bacteria contain two ketosynthase domains and catalyse repeated condensation of the starter *p*-coumaric acid (4-hydroxy-cinnamic acid) with three malonyl-CoA elongation units to form a tetraketide. In recent years, several chalcone synthases similar to that involved in naringenin biosynthesis have been found in other actinobacteria. This raises the question of whether naringenin is produced by more actinobacteria and if the pathway is identical to that of plants. This article aims to analyse the analogies and differences between the naringenin CHS and other naringenin biosynthetic enzymes in plants and actinobacteria. We summarize here evidence showing that several actinobacteria using type III polyketide synthases produce a variety of metabolites including pigments, antibiotics and antitumor agents. Several of these microorganisms use malonyl-CoA as a starter unit instead of a phenylpropanoid starter, and some of them are able to cyclize the polyketide chain to form aromatic intermediates.

The content of this article is organized according to the sequence of reactions that take place in the biosynthesis of naringenin and related chalcones. Following the Introduction, we study the chalcone synthases in plants and actinobacteria (Section 2 and Section 3). Then we describe the molecular genetics and biochemistry of reactions involved in the biosynthesis of the chalcone starter/substrate molecules including aromatic-derived and linear polyketides (Section 4). Later, we review the activation of these substrates by CoA ligases (Section 5). We include in Section 6 the biosynthesis of non-proteinogenic amino acids that form part of vancomycin-type of antibiotics, formed by enzymes related to chalcone synthases. Finally, we include the biotechnological processes to produce naringenin and derived compounds (Section 7).

## 2. Biosynthesis of (2S)-Naringenin in Plants

Biosynthesis of naringenin in plants is well known to proceed through the phenylpropanoid pathway [24]. In plants, this pathway involves five enzymes, namely phenylalanine/tyrosine ammonia lyase, trans-cinnamic acid 4-hydroxylase (when phenylalanine is used as precursor), *p*-coumaroyl-CoA ligase, chalcone synthase, and chalcone isomerase (Figure 2).

The naringenin biosynthesis pathway in plants starts with the formation of *p*-coumaroyl-CoA, which is synthesized from L-tyrosine or from L-phenylalanine by the action of a phenylalanine/tyrosine ammonia lyase (PAL/TAL) that converts one of these precursor amino acids into *p*-coumaric acid. Monocot plants use both amino acids as precursor of *p*-coumaroyl-CoA, whereas dicots only use phenylalanine that requires in these plants a transcinnamoyl-4-hydroxylase to introduce a hydroxyl group at position 4 of the phenylalanine-derived cinnamic acid (Figure 2). A central role in the naringenin synthesis is played by the naringenin chalcone synthase (EC 2.3.1.74). Plants usually have several genes encoding isoforms of chalcone synthases. These enzymes, despite their simplicity show a notable functional promiscuity in the utilization of different substrates [25].

The naringenin chalcone intermediate in plants is cyclized to form naringenin by the combined action of the chalcone synthase and the chalcone isomerase (CHI). First, the tetraketide formed by the CHS is cyclized by the same enzyme to form 2,4,4,6 tetrahydroxychalcone. The last step in naringenin biosynthesis is performed by the CHI that catalyses a stereospecific isomerization resulting in the cyclization of the former intermediate into a flavanone [26,27]. Phylogenetic studies of CHIs revealed that there are two major types of CHIs in plants; an additional third group includes chalcone-isomerase-like proteins, that increase chalcone formation by interacting closely with the chalcone synthases avoiding the derailment of the polyketide that would form side products [26,28]. Therefore, this chalcone-isomerase-like activity increases the efficiency of synthesis of selected chalcones. A specific molecular mechanism has been proposed for the naringenin chalcone isomerization based on structural studies of CHI enzymes. After positioning of the chalcone substrate in the CHI active center, a transient enol is formed from the αβ-unsaturated double bond of the chalcone substrate. The subsequent deprotonization of the enoyl intermediate leading to the formation of the naringenin pyrone ring is pH-dependent [27,29]. Interestingly, this naringenin chalcone in plants is able to self-cyclize slowly under acidic conditions without the requirement of the chalcone isomerase; therefore, the last enzyme might be dispensable when the biosynthesis of naringenin takes place under acidic conditions [30].

## 3. The Naringenin Chalcone Synthase of *S. clavuligerus*: Comparison with Other Bacterial Chalcone Synthases

In *S. clavuligerus,* as occurs in plants, the naringenin chalcone synthase (Ncs) catalyses the condensation of tyrosine-derived *p*-coumaric acid and three malonyl-CoA units to form the naringenin chalcone (Figure 2). The role of the *S. clavuligerus* chalcone synthase encoding gene (*ncs*) has been stablished by gene deletion and complementation of the deleted mutant [21]. This gene is present in a single copy in the genome of this actinobacteria and encodes a CHS of 351 amino acids. The bacterial CHSs are always smaller than those of plants [16,31]. Comparative analysis of *S. clavuligerus ncs* encoded protein with the homologous enzymes of plants reveal an average identity of 29.3%, although they play a similar catalytic function (Figure 3).

### 3.1. Bacterial Chalcone Synthases That Use Aromatic or Aliphatic Starter Units

Two classes of chalcones are formed by the bacterial chalcone synthases according to the starter units utilized, that in some cases are CoA-activated aromatic acids and in others are malonyl-CoA units. The final products of the CHSs that use aromatic acid units depends on the starter unit used by the condensing enzyme such as benzoic acid in enterocins or caffeic acid in saccharomicins A and B [32,33]. The CHS of *Streptomyces griseus, Saccharopolyspora erythraea* and *Streptomyces coelicolor* A3(2) is encoded by the *rppA* gene, so named for red-brown pigment production [34,35,36,37]. The RppA chalcone synthase differs from *S. clavuligerus* Ncs, in that it uses malonyl-CoA as starter unit instead of the *p*-coumaroyl-CoA unit.

RppA condenses five malonyl-CoA units (one starter and four elongation units) to form a pentaketide intermediate and then 1,3,6,8-tetrahydroxynaphtalene (THN), a precursor of the 1,4,6,7,9,12-hexa-hydroxy-perylene 3,10 quinone (HPQ), which autopolymerizes to form bacterial HPQ melanin [38]. The THN intermediate may also be oxidized to flaviolin (2,5,7-trihydroxy-1,4-naphtoquinone) [39] (Figure 4). *S. clavuligerus* does notproduce red brown pigment similar to that formed by the RppA chalcone synthase. Conversely, *S. coelicolor* RppA does not synthesize naringenin, which is produced by a small group of species closely related to *S. clavuligerus*, such as *Streptomyces jumonjinensis* [21,40]. A gene encoding a naringenin chalcone synthase has also been found in *Streptomyces katsuharamanus*, but there is no report of production of naringenin by this strain. Moreover, when *S. coelicolor* cultures were supplemented with *p*-coumaric acid or when the strain was transformed with the *S. clavuligerus tal* gene (that account for the formation of *p*-coumaric acid), the transformants were unable to produce naringenin [21] indicating that the RppA chalcone synthase is unable to accept the *p*-coumaric acid starter unit or that *S. coelicolor* lacks an adequate *p*-coumaroyl-CoA ligase to activate this precursor (Figure 2). Noteworthy, the amino acid sequence of *S. clavuligerus* Ncs is largely conserved in relation to the RppA chalcone synthases of *S. coelicolor* (68% identity) and *S. avermitilis* (75% identity) indicate that *S. clavuligerus* Ncs is closely related phylogenetically to the RppA chalcone synthases of other *Streptomyces* species, forming a subfamily of chalcone synthases [21] (Figure 3). The chalcone synthases phylogenetic three shows a distant relatedness between plants and actinobacterial CHSs, which excludes the possibility of a recent horizontal gene transfer between them. Alignment of the amino acid sequences of *S. clavuligerus* Ncs with the RppA proteins of other *Streptomyces* species allowed us to identify the catalytic triad C**^138^**H**^270^**N**^303^** described in plant chalcone synthases, and other amino acids lining the active site pocket (**^106^**CT**^107^**, C**^171^**, F**^188^**, F**^233^**, A**^305^**), also conserved in plant chalcone synthases [18].

The available evidence indicates that the cyclization of the polyketide chain by Ncs that results in the formation of the two rings of naringenin is different from that performed by the RppA chalcone synthase that leads to the aromatization and formation of the required THN intermediate. In the red-brown pigment, this cyclization of the pentaketide chain proceeds through a carbon-carbon condensation of the carboxyl group at the start of the polyketide chain with an CH_2_ at the end of the pentaketide, simultaneous with a decarboxylation (Figure 4) [41].

Initial crystallographic studies of some CHSs reveal an important role of the size of the entry channel to the internal cavity in the selection of the chalcone starting unit [42,43]. The different size of the internal cavity explains the final products of the CHSs, e.g., the chalcones formed from aromatic starters or the tetrahydroxynaphtalene formed from malonyl-CoA. The *Streptomyces* RppA CHSs were proposed to have a narrow entry channel that allows only the entry of the malonyl starter unit [41], whereas the plant, *S. clavuligerus* Ncs and other actinobacteria “aromatic chalcone synthases” allow the entry of bulky starter units such as *p*-coumaric-CoA, caffeoyl-CoA or benzoyl-CoA. It was initially proposed that the tyrosine Y^224^ residue in the entry channel of the internal cavity of the RppA chalcone synthase was a determinant for the selective entry of malonyl-CoA as a starter unit; however, further analysis of mutants in which the Y^224^ was replaced by other amino acids disputed this finding [37]. Therefore, the mechanism that selects the starter unit is more complex and is still unclear [38,41].

### 3.2. Role of the P_450_ Monooxygenase (NcyP) Encoded by a Gene Linked to ncs

Genes encoding P**_450_** monooxygenases are frequently associated with genes for type III polyketide synthases. Adjacent to the *ncs* gene in *S. clavuligerus*, there is a gene encoding a P**_450_** monooxygenase (NcyP) of 454 amino acids which is transcribed separately from *ncs*. The NcyP monooxygenase is required for the biosynthesis of naringenin as shown by gene deletion and complementation of the *ncyP* mutant [21]. This P**_450_** monooxygenase appears to be involved in the cyclization of the naringenin chalcone to form the pyrone ring of naringenin (Figure 2). No gene encoding an enzyme similar to the plants naringenin chalcone isomerase was found in the *S. clavuligerus* genome, suggesting that the final step in the cyclization of the naringenin structure is performed by a different enzymatic system. This information is consistent with the observation that the naringenin chalcone isomerase in plants is conditionally dispensable, as indicated above. Multiple P**_450_** monooxygenases are encoded in the genome of *S. clavuligerus* with identity percentages to NcyP ranging from 35 to 47%. However, the NcyP monooxygenase is essential for the biosynthesis of naringenin since other P**_450_** oxygenases present in *S. clavuligerus* are unable to replace this enzyme function in the *ncyP*-deleted mutant. It is likely that only the NcyP monooxygenase is able to form a functional two-protein complex with the Ncs protein. Genes homologous to *ncs*-*ncyP,* with a similar linked arrangement, occur in other actinobacteria; all the cytochrome P**_450_** encoding genes shown in Figure 5 have an adjacent gene for a chalcone synthase, except in *S. noursei* and *S. natalensis.* A high identity percentage of the cytochrome P**_450_** encoded proteins was observed in two clavulanic acid producers, namely *S. jumonjinensis* and *S. katsuharamanus,* and in *Streptomyces inhibens,* with average identities of 83.7 and 76.3% for Ncs and NcyP, respectively. However, the highest identities were observed in species of *Kitasaspora* and *Saccharothrix*, e.g., in *Kitasaspora aureofaciens* where the *ncs-ncyP* genes encode proteins with 84 and 80% amino acid identity and in *Saccharotrix* ST-888 with a Ncs-NcyP protein identity of 87% and 80%, respectively, to those of *S. clavuligerus* (Figure 3 and Figure 5), although it is not known whether *Kitasaspora* or *Saccharotrix* species produce naringenin. In other *Streptomyces* species, in which the homologous genes, *rppA* and *rppB* are involved in red-brown pigment production, as *S. coelicolor, S. griseus* or *S. venezuelae*, the identity of these proteins is lower, with identities to Ncs and NcyP in the actinobacteria *Sacc. erythraea* of 75 and 41%, respectively [36]. The *rppA* and *rppB* genes are adjacent and have been proposed to have a role in the crosslinking of TNH molecules and in the oligomerization of flaviolin. The low similarity of NcyP of *S. clavuligerus* to the RppB proteins of *Sacc. erythraea, S. coelicolor, S. venezuelae* or *S. avermitilis* suggests that these P**_450_** oxygenases have different roles in the late cyclization steps of their respective chalcones (Figure 5).

The *rppB* gene, associated to *rppA*, is required for the biosynthesis of the THN intermediate, and the encoded P**_450_** oxygenase may serve as an enzyme for the cyclization of the pentaketide chain to form THN. In addition, it has been proposed that this P**_450_** in *S. coelicolor* is involved in the dimerization of the THN intermediate, leading to the formation of flaviolin, a quinone molecule derived from THN, and to further oligomerization of this compound [36,37]. Similarly, *Streptomyces toxytricini* NRRL 15,443 contains a cluster formed by a chalcone synthase gene and two adjacent genes encoding P**_450_** monooxygenases. The two encoded P**_450_** monooxygenases, indistinctly, oxidize flaviolin to form oligomers in vitro [39]. A fourth gene in the cluster, *stmo,* encoding an additional quinone-forming oxygenase with a cupin fold in its structure, is involved in the oxidation of THN to flaviolin. This gene cluster is involved in the biosynthesis of bacterial melanin and, when expressed in *E. coli,* protects the cells from UV radiation [39]. 

We have found, in *S. coelicolor*, downstream of *rppB*, a *stmo* orthologous gene, which encodes a protein 77% identical to *S. toxytricini* Stmo, which has not yet been characterized functionally. The Stmo protein of *S. coelicolor* may be used as a catalyst for the formation of flaviolin as described in *S. toxytricini.* Since *stmo* genes are associated with the *rppA-rppB* gene cluster in several *Streptomyces* species known to synthesize flaviolin and red-brown pigment, it seems that this quinone forming oxygenase is essential for the biosynthesis of these compounds. However, there is no *stmo* homologous gene in the genome of *S. clavuligerus*, in agreement with the lack of formation of flaviolin by this strain.

## 4. Biosynthesis of *p*-Coumaric Acid and Other Related Starter Molecules: Role of Ammonia Lyases

As indicated above, different CHSs use distinct starter and elongation units to form the chalcone intermediate.

### 4.1. Biosynthesis of p-Coumaric Acid in Plants and Yeasts

In plants, the *p*-coumaric acid started is formed either from tyrosine or phenylalanine by a tyrosine (TAL) or phenylalanine (PAL) ammonia lyase [44]. The PAL and TAL enzymes are very similar, but depending on their biological source they show preference for phenylalanine over tyrosine ranging from 1 to 600.000-fold [45,46].

The characterized plant PAL ammonia lyases are usually homotetramers. Their native form size ranges from 300 to 340 kDa [47]. In several plants, the PAL enzymes are encoded by multigene families, ranging from 4 to 8 copies per genome [48,49,50,51]. The PAL enzymes of several monocot plants have been shown to have both PAL and TAL activities, including the monocot *Zea mays* PAL1 [45] and the recently purified PAL2 [51], which has a higher PAL than TAL activity when the encoding gene is expressed in *E. coli*. The crystal structure of one of the eight *Sorghum* PAL ammonia lyase isoforms enabled definition of the amino acid residues that are important in the utilization of phenylalanine or tyrosine as a substrate [52]. Additionly, in the red yeast *Rhodosporidium toruloides*, the ammonia lyase recognizes both aromatic amino acids [53] as shown by heterologous expression in *S. cerevisiae*.

### 4.2. Bacterial Amino Acid Ammonia Lyases

Several examples of amino acid ammonia lyases have been described in bacteria in the last decades. The best-known are the histidine ammonia lyases (HAL) and the tyrosine ammonia lyases [44].

The TAL, PAL and HAL enzymes perform a Friedel-Crafts deamination reaction that results in the formation of α, β-unsaturated aryl-propanoid acids [54]. A histidine ammonia lyase that forms trans-urocanic acid from histidine was initially described in *Pseudomonas putida* [55]. Later, Louie et al. [44], using the HAL enzyme of *Rhodobacter sphaeroides* (now *Cereibacter sphaericus*), proved that this enzyme was able to use tyrosine, phenylalanine and histidine as substrates and, therefore, the enzymes annotated as HAL are likely broad spectrum aromatic amino acid/histidine ammonia lyases. Aromatic ammonia lyases are present in diverse bacterial classes including actinobacteria as *Streptomyces maritimus,* photosynthetic bacteria as *R. sphaeroides*, or myxobacteria as *Sorangium* sp. [56,57,58,59]. The photobacteria *Rhodobacter capsulatus* has a TAL enzyme involved in the formation of its yellow enzyme chromophore [60,61]. This TAL enzyme has high selectivity for tyrosine over phenylalanine (150-fold) and is considered the prototype of tyrosine ammonia lyases [60].

Crystalographic studies of the *R. sphaeroides* TAL reveal that the His^89^ residue of this enzyme forms hydrogen bonds with the 4-hydroxyl group of the tyrosine substrate. Whereas the TAL ammonia lyases contain a His^89^ residue, the PAL and HAL enzymes have a phenylalanine at this position. The replacement of His^89^ by Phe^89^ in the *R. sphaeroides* ammonia lyase change the substrate specificity in favour to phenylalanine [44].

The availability of bacterial TAL enzymes lacking phenylalanine ammonia lyase activity was used by Nishiyama et al. [62] to direct the biosynthesis of phenylpropanoids in plants; when the *R. sphaeroides tal* gene was introduced in *A. thaliana*, its expression led to the formation in early development stages of large amounts of anthocyanins and production, in later steps, of quercetin glycosides. This is a good example of the possible use of aromatic ammonia-lyases with different substrate specificity to modify the production of flavonoids, anthocyanins and other products in plants.

In *S. clavuligerus,* as well as in all plants, the starter unit for naringenin biosynthesis is *p*-coumaric acid (4-hydroxy-cinnamic acid), which in this actinobacterium derives from L-tyrosine [21]. In *S. clavuligerus,* the TAL enzyme has 559 amino acids and is encoded by the gene SCLAV_5457 (*tal*). Cloning, deletion and complementation of the *S. clavuligerus tal* gene showed that the encoded enzyme is essential for the biosynthesis of naringenin [21].

Comparison of the TAL enzymes of *S. clavuligerus, Saccharothrix espanaensis* and other actinobacteria and the model TAL enzyme of *R. sphaeroides* shows that the His**^89^** residue in the *R. sphaeroides* TAL corresponds to Tyr**^105^** in *S. clavuligerus* and closely related *Streptomyces* species. The available results indicate that the ammonia lyase of *S. clavuligerus* is a true TAL; this conclusion is supported by the lack of a 4-cinnamoyl-hydroxylase gene in the *S. clavuligerus* genome that would contribute to form coumaric acid from phenylalanine. In vitro studies with the *Streptomyces globisporus* aminomutase that converts α-tyrosine to β-tyrosine, involved in the synthesis of (*S*)-3-chloro-4,5-dihydroxy-β-phenylalanine component of the enediyne C-1027 antitumor agent, showed that this enzyme is also a TAL, since the in vitro product of the first reaction step is *p*-coumaric acid [63].

### 4.3. The Aromatic Ammonia Lyases and the Aminomutases Contain a Methylidene Imidazol-5-one (MIO) Prostetic Group

The MIO structure works as a prosthetic group and is formed by autocatalytic cyclization and dehydration of the tripeptide Ala-Ser-Gly that forms part of the amino acid sequence of ammonia lyases and aminomutases [64]. In the *S. clavuligerus* TAL, this tripeptide corresponds to amino acids **^164^**ASG**^166^** [21]. The mechanism of ammonium elimination by the aromatic ammonia lyases is well established; the highly electrophilic MIO group is able to stereo-specifically abstract a proton from the β-CH**_2_** of the substrate L-aromatic amino acid resulting in the removal of the amino group [46,65]. The aminomutases that form β-amino acids have a similar mechanism. They also contain a MIO prosthetic group, but in the reactions catalysed by these enzymes, the overall conversion has a second half- reaction involving the transfer of the amino group from the α to the β position, leading to the formation of the rare beta amino acids instead of the deamination of the L-amino acid [63,66,67].

### 4.4. Enzymes for the Conversion of Trans-Cinnamic Acid and p-Coumaric Acid into Caffeic Acid and Ferulic Acid

Caffeic acid and ferulic acid are intermediates in the formation of flavonoids and lignin and have important applications in the food industries. Caffeic acid has antioxidant, anti-inflammatory, antitumor activities and has an effect on atherosclerosis; ferulic acid is used as a component of nutraceuticals and other functional foods. In addition, some bacteria synthesize either caffeic acid or ferulic acid as components of secondary metabolites, as occurs in the biosynthesis of the antibiotics saccharomicins A and B (Figure 6A) [33]. The saccharomicins are complex oligosaccharides produced by *Sac. espanaensis* that contain an aglycon moiety of caffeoyl-taurine. These antibiotics are potent antibacterial compounds active against Gram positive bacteria including *Staphylococcus* and vancomycin-resistant *Streptococcus* strains. Enzymes occurring in *Sac. espanaensis* and other bacteria perform additional hydroxylation and methylations on *p*-coumaric acid to form caffeic acid. The conversion of the *p*-coumaric acid to its 3-hydroxyderivative is catalysed by a coumarate 3-hydroxylase encoded by a gene that has been cloned from *Sac. espanaensis* [33] and also from plants [68,69,70].

In plants, the modifications by late oxygenases of either cinnamic acid or *p*-coumaric acid occur after their activation by formation of shikimate or quinate esters [71,72,73]. A similar *p*-coumaroyl 3-hydroxylase that introduced the 3-hydroxyl group in ester activated *p*-coumaric acid has been found in *A. thaliana* [74,75].

In contrast to what occurs in plants, studies on the biosynthesis of caffeic acid in *Sac. espanaensis* showed that the *p*-coumaric acid is converted to its 3-hydroxy-derivative by a 3-oxygenase encoded by the *sam5* gene without previous esterification [33]. This is an important difference between the plant conversion of *p*-coumaric acid to caffeic acid and that observed in *Sac. espanaensis*, although further research on the characterization and crystallization of these enzyme of plants and *Sac. espanaensis* has to be performed to clarify their substrate specificity.

Bioconversion studies using *Streptomyces faerulens* isolated from soil by phenolic compounds enrichment techniques have shown that *p*-coumaric acid is transformed into caffeic acid and also to the novel product 4-hydroxybenzoic acid, in which the three carbons side chain has been shortened to the C-1carboxylic acid. *p*-Coumaric acid is abundantly produced in plants and may be converted to other aromatic compounds by industrial processes [76].

Structurally related to caffeic acid is the ferulic acid (3-methoxy-coumaric acid) synthesized by methylation of a hydroxyl group in the former compound (Figure 1). The gene encoding the caffeic acid O-methyltransferase (O-MT) has been cloned from several plants, including *A. thaliana* and *M. sativa* [77,78]; recently, the caffeic acid O-methyl transferase from *Azadirachta indica*, an aromatic plant (Neem) that produces a variety of phenolic compounds, has been characterized. The enzyme expressed heterologously in *E. coli* has been identified as a S-adenosylmethionine (SAM)-dependent methyl transferase and has been proposed to be used for the large-scale production of ferulic acid [79]. The *A. thaliana* caffeoyl-O-MT has been found to methylate serotonin (5-hydroxytryptamine), which offers new possibilities to methylate other molecules of pharmacological interest [76].

In contrast to the many reports on ferulic acid biosynthesis in plants, there is little information on the molecular mechanisms of the enzymatic steps for the conversion of cinnamic acid to caffeic acid or to ferulic acid in actinobacteria. Genes encoding caffeoyl O-methyltransferases are present in the genomes of different *Streptomyces*. In *Streptomyces avermitilis,* a SAM-dependent methyltransferase, encoded by SAV_OMT5, was found to methylate caffeic acid and caffeoyl-CoA to ferulic acid and feruloyl-CoA, respectively. Heterologous expression of the encoding gene and purification of the methyl transferase indicate that this enzyme was also able to methylate other flavones such as 2,3 dioxyflavone, 3,4 dioxyflavone, 6,7 dioxyflavone and quercetin [80]. Zhang et al. [81] found nine O-MTs in a bioinformatic study of *Streptomyces virginiae* genome, and later characterized three of them, namely O-MT02, O-MT03 and O-MT06. Of these three methyltransferases, O-MT03 and O-MT06 are able to convert caffeic acid into ferulic acid. The amino acids of the methyl transferase involved in the recognition of the substrates caffeic acid and SAM were identified and a mechanism for the methyl group transfer was proposed [81]. In addition, the actinobacteria may obtain ferulic acid by degradation of plants lignin using different types of enzymes, including feruloyl esterases.

## 5. Activation of *p*-Coumaric Acid and Related Chalcone Precursors by aryl-CoA Ligases

The phenylpropanoid CHSs require the activation of trans-cinnamic acid, p-coumaric acid, benzoic acid, caffeic acid, ferulic acid and sinapic acid (3,5 dimethoxy-coumaric acid, Figure 1) to their corresponding CoA derivatives that are used as starter units. A putative *p*-coumaroyl-CoA ligase was found in the genome of *S. clavuligerus*, corresponding to the gene SCLAV_ 3408. This gene encodes a CoA ligase containing an ATP binding site (**^386^**TGDIL**^390^**) that is used in a first activation of *p*-coumaric acid to form *p*-coumaroyl-AMP, which is subsequently converted to *p*-coumaroyl-CoA (second half-reaction) by the same enzyme (Figure 2). Enzymes similar to the *S. clavuligerus p*-coumaroyl-CoA ligase have been found in the naringenin producer *S. jumonjinensis* (74% identity), in *S. coelicolor* (69% identity) for *p*-coumaroyl activation, in *Sac. espanaensis* (54% identity) for the activation of caffeic acid, and in *Streptomyces maritimus* (31% identity) for the activation of benzoic acid. The different degree of conservation of the amino acid sequences in the *p*-coumaroyl CoA ligase, caffeoyl-CoA ligase and benzoyl-CoA ligase indicates that these enzymes have evolved to adapt to the distinct substrates, although there are no detailed studies of their substrate specificity.

### 5.1. Substrate Specificity of p-Coumaroyl-CoA Ligases in Plants and Filamentous Fungi

Several *p*-coumaroyl-CoA ligases (4CL) isoforms have been found in plants with slightly different substrate specificity [82]. In order to determine the substrate specificity of each isoform, Schneider et al. [83] isolated three of the four isoforms present in *A. thaliana*. Isoforms At4CL1 and At4CL3 behave as typical *p*-coumaroyl-CoA ligases that use *p*-coumaric acid efficiently and caffeic acid, ferulic acid and sinapic acid less effectively as substrates. After purification, these isoforms utilize cinnamic acid very poorly, while sinapic acid was not activated at all [84]. The isoform 4CL2 prefers caffeic acid as substrate rather than *p*-coumaric acid and does not activate ferulic acid. Further studies on the amino acid sequence of different 4CL proteins determined the presence of twelve, non-consecutive, amino acid residues lining the substrate binding pocket of the *p*-coumaroyl-CoA ligases [83]. Mutation of some of these residues results in a gain-of-function and some of the mutant enzymes efficiently recognize ferulic acid and sinapic acid as substrates, in contrast to the wild-type parental strain. Similarly, deletion of one of these amino acids in the soybean 4CL4 *p*-coumaroyl-CoA ligase resulted in a mutant able to activate sinapic acid [85]. Alignment of the *S. clavuligerus* 4CL with the *A. thaliana* enzymes reveals that 5 of the 12 amino acids lining the *p*-coumaric acid recognition pocket are identical, and some additional amino acids are functionally conserved.

Several 4CL isoforms are encoded in ascomycetes and other filamentous fungi genomes. Eight *p*-coumaroyl CoA ligase related enzymes have been found in *Penicillium chrysogenum*; two of them, named PhlA and PhlB, have been characterized by in vivo molecular genetics and by in vitro enzyme studies as phenylacetyl-CoA ligases for the formation of benzylpenicillin [86,87], and a third, PhlC, activates adipic acid [88,89]; the remaining isoforms are annotated as *p*-coumaroyl-CoA ligases [90].

### 5.2. Activation of Benzoic Acid, a Rare Aromatic Precursor in the Biosynthesis of the Polyketide Enterocins in Streptomyces Maritimus

During a search in marine environments for novel antibiotics, a large gene cluster (21.3 kb) was cloned from *S. maritimus* [91]. Heterologous expression in *Streptomyces lividans* and HPLC-mass spectrometry studies of the transformant fermentation products confirmed that this cluster encodes the biosynthesis of the polyketides enterocins and wailupemycins [20]. The formation of these polyketides is performed by an atypical chalcone synthase that uses benzoyl-CoA as starter unit and condenses it with seven malonyl-CoA units, forming the octaketide intermediate. Moore et al. [31] proposed that the benzoyl-CoA is formed from phenylalanine by a PAL enzyme encoded by the *encP* gene. This was confirmed by heterologous expression of *encP* in *S. coelicolor*, which resulted in formation of cinnamic acid. In addition to the type III PKS, the enterocin cluster contains the genes *encC* and *encD* for an ACP (acyl-carrier protein) and a ketoreductase, but in contrast to type II polyketide synthases gene clusters, it lacks genes encoding cyclases and aromatases.

The origin of the benzoic acid starter of enterocins and its activation to benzoyl-CoA is intriguing. Labelled precursors studies showed that only the benzoic acid unit but not the complete three carbon side chain is incorporated in enterocins; in other words, the side chain of cinnamoyl-CoA is modified by an enoyl-CoA hydratase (EncI) and further cleaved by β-oxidation reactions to form benzoyl-CoA (Figure 6B).

Inactivation of the phenylalanine ammonia lyase *encP* gene in *S. maritimus* resulted in a lack of production of enterocins, which was reverted by supplementation with cinnamic acid or benzoic acid, or by transformation with the *encP* gene. Complementation of enterocins production with exogenous benzoic acid in the *encP* mutant requires a functional benzoyl-CoA ligase (*encN*) gene. Knock-out of the *encN* gene does not prevent enterocin formation, but results in lack of utilization of exogenous benzoic acid (Figure 6B).

Disruption of the fatty acid catabolism-like genes *encH, I* or *J*, revealed that these three genes are involved in enterocin biosynthesis but they are not essential, indicating that they may be replaced by other fatty acid catabolism genes present elsewhere in the genome [32,92]. These authors propose that shortening of the three carbons side chain of the cinnamic acid takes place by a process similar to that of the α-oxidation of fatty acids, although the cluster lack an β-hydroxyacyl-CoA dehydrogenase gene required to complete the shortening of the side chain; alternatively, the oxidation of the three carbons side chain may be completed by a retro-aldol reaction leading to the formation of benzoyl-CoA. This alternative proposal is supported by the observation that mutants disrupted in *encN* are still able to produce enterocins, suggesting that the shortening of the three carbon side chain proceeds by the alternative retro-aldol pathway. The moderate conservation of the amino acid sequence (27% identity) of the *S. maritimus* benzoyl-CoA ligase and the *S. clavuligerus p*-coumaroyl-CoA ligase indicates that these two enzymes belong to phylogenetically distant branches of the aryl-CoA ligases family.

## 6. Biosynthesis of 4-Hydroxyphenylglycine and (S) 3,5 Dihydroxyphenylglycine, Related to Aromatic Starter Units of Chalcones

Connected to the formation of the phenylpropanoid starter units in bacteria is the biosynthesis of 4-hydroxyphenylglycine (HPG) and (S)3,5 dihydroxyphenylglycine (DPG), two non-proteinogenic amino acid components of several non-ribosomal peptide antibiotics including vancomycin, teicoplanin, chloroeremomycin, norcadicins, complestatin, and the calcium-dependent antibiotic CDA, among others [93]. Both HPG and DPG play a key role in the maintenance of the rigid structure of the heptapeptide chain of the vancomycin-like antibiotics and are a site for late glycosylations. These two non-proteinogenic amino acids occur together in some vancomycin-type antibiotics, but not in all of them, and are encoded by two separated clusters containing four genes each; in the case of HPG, this compound is synthesized from prephenate or tyrosine, while surprisingly, the DPG is formed by a type III polyketide synthase.

### 6.1. Biosynthesis of 4-Hydroxyphenylglycine (HPG)

Studies on the eremomycin gene cluster revealed the enzymes for HPG biosynthesis [93]. In *Amycolatopsis orientalis*, the first enzyme is the prephenate dehydrogenase (PDH), that converts prephenate to 4-hydroxyphenylpyruvate. The second enzyme, encoded by *hpgS*, is a Fe^2+^-dependent decarboxylating oxygenase (so called hydroxymandelate synthase, HMS) that converts 4-hydroxyphenylpyruvate to 4-hydroxymandelate [94]. The third enzyme, a hydroxymandelate oxidase (HMO) [95] encoded by *hpgO,* transforms 4-hydroxymandelate to 4-hydroxyphenylformate, which is finally converted to 4-hydroxyphenylglycine by a transaminase encoded by *hpgT* that uses L-tyrosine as an amino group donor (Figure 7). In the transamination reaction, tyrosine is converted to 4-hydroxyphenylpyruvate that enters in a cycle forming 4-hydroxyphenylglycine again using the three last enzymes of the pathway. This explains the early finding of incorporation of labelled ^13^C-tyrosine into 4-hydroxyphenylglycine [96,97]. The initial enzymatic step that converts prephenate to 4-hydroxyphenylpyruvate serves to prime the reaction cascade, providing a basal level of p-hydroxyphenylpyruvate to initiate the three enzyme cycle. The similarity of these enzymes with other enzymes of bacterial metabolism has been reviewed by Toma et al. [98].

### 6.2. Biosynthesis of 3,5-Dihydroxyphenylglycine (DPG)

Regarding the second non-proteinogenic amino acid 3,5-dihydroxyphenylglycine, studies on the biosynthesis of the antibiotic balhymicin in *Amycolatopsis mediterranei* [99] and on the related eremomycin gene cluster of *Amycolatopsis orientalis* show that DPG is formed through a polyketide pathway [100] but not from a tyrosine precursor, in contrast to what occurs in the biosynthesis of HPG. The DPG biosynthesis enzymes are encoded by four genes named *dpgABCD*. Disruption of these genes prevented the formation of the DPG moiety of these vancomycin-type antibiotics and the disrupted mutants can be complemented with 3,5 dihydroxyphenyl acetic acid [99], suggesting that this compound is an intermediate of the DPG biosynthesis. The *dpgA* gene encodes a type III PKS that uses four units of malonyl-CoA (one starter and three elongation units); this enzyme is a chalcone synthase that has only 22% identity to *S. clavuligerus* Ncs. The tetraketide formed is cyclized/aromatized by the same enzyme, resulting in the formation of dihydroxyphenylacetic acid [101]. The efficiency of formation of dihydroxyphenylacetic acid is increased 35-fold by the action of the enzymes DpgB and DpgD; this intermediate is converted to dihydroxyphenylglyoxilic acid by the enzyme encoded by *dpgC* [100,101] (Figure 8). Heterologous expression of the *dpgA* in *S. lividans* led to the formation of dihydroxyphenylglyoxylic acid; this compound is converted to dihydroxyphenylglycine by a transamination reaction catalyzed by the transaminase HpgT described above [93]. Expression in *E. coli* of the *dpgABCD* genes and purification of the encoded proteins confirmed the conversion of dihydroxyphenylacetic acid to 3,5 dihydroxyphenylglyoxylic acid which is transformed in dihydroxyphenylglycine by a transamination reaction. In summary, this information supports the proposed pathway for production of DPG using a type III PKS and a transaminase [100]. Noteworthy, this transaminase is shared in the pathway of both HPG and DPG. A provocative result of this transamination is the formation of 4-hydroxyphenylpyruvate from tyrosine in the DPG pathway that may serve to increase the production of HPG as described above, supporting a positive cooperation of the biosynthesis of HPG and DPG in those actinobacteria that contain both vancomycin-type antibiotic gene clusters.

## 7. Heterologous Production of Naringenin in Yeasts: Biotechnological Applications

Heterologous expression of genes for naringenin biosynthesis has been achieved in *Escherichia coli* and more efficiently in yeasts. The use of yeasts as hosts has the advantage of efficient plant genes expression, adequate posttranslational modification of plant proteins and the presence in yeast cells of intracellular compartments (e.g., peroxisomes, vacuoles and traffic vesicles) in the case that they are required for proper localization of the enzymes of the pathway. The naringenin biosynthesis genes of *A. thaliana* and other plants have been used in many yeast expression studies. These clusters require little modifications to be expressed in the yeasts *Saccharomyces cerevisiae* and *Yarrowia lipolytica*. The main problem for the production of naringenin in yeasts is the absence in these organisms of the *p*-coumaric acid biosynthetic step. The lack of formation of the naringenin precursor has been bypassed by feeding *p*-coumaric acid to the cultures.

Increased levels of naringenin production have been achieved in yeast by improving the formation of acetyl-CoA and malonyl-CoA [102,103,104], or enhancing the β-oxidation pathway for the production of polyketide intermediates [103,104,105]. Modification of the negative regulation of the shikimate pathway, e.g., by removal of the feedback regulation of tyrosine formation at the DAHP level and obtention of phenylpyruvate decarboxylase mutants, to save the phenylpropionate pools, also resulted in higher levels of naringenin production [104,105].

Other strategies have been used to improve steps which are bottlenecks in the pathway, such as the reactions catalysed by the naringenin chalcone synthase, or by a cytochrome P**_450_** monooxygenase, which putatively enhances the cinnamoyl-4-hydroxylase activity [104]. This was achieved by modifying the promoters to enhance the induction by xylose of the expression of the genes [106,107], or by increasing the genes copy number [108,109,110]. The best naringenin production reached 1210 mg/L in a bioreactor after supplementing the cultures with *p*-coumaric acid [109].

The introduction of additional genes in yeasts together with those required for naringenin biosynthesis results in the formation of compounds derived from naringenin (Figure 9). These genes encode 3′-hydroxylases from plant origin to produce 2S-eriodictyol, glycosyl transferases to form naringenin glycosides, plant prenyltransferases to produce 8-prenylnaringenin, or other enzymes to form dihydromyricetin or 6 and 8 hydroxynaringenin (Figure 9).

## 8. Conclusions and Future Outlook

A significant amount of information has been accumulated in the last two decades on the molecular mechanisms of naringenin and other phenylpropanoids biosynthesis in actinobacteria. Comparative studies with the biosynthesis of naringenin in plants reveal that *S. clavuligerus* is able to produce naringenin from tyrosine using a naringenin chalcone synthase that utilizes *p*-coumaric acid as starter unit. Numerous secondary metabolites including pigments, antibiotics or antitumor agents are synthesized by type III PKSs related to the naringenin chalcone synthase; however, in contrast to the formation of naringenin chalcone, many of these type III PKSs in actinobacteria are involved in the biosynthesis of THM, flaviolins and bacterial melanin which derive from a pentaketide formed for a malonyl-CoA starter. This opens the possibility to obtain novel molecules with different biological activities by combining starters and elongation units from distinct actinobacterial gene clusters. This strategy may be used to obtain novel antibiotics, e.g., antibacterials of the vancomycin type or new antitumor agents of the enediyne class by modification of the aromatic pathway and/or the polyketide synthase. In the last few years, numerous reports have been published on the production of naringenin and related compounds in yeast taking advantage of the good efficiency to express plant genes in these eukaryotic microorganisms. This has resulted in strains with a high capability to produce naringenin, and the same may apply to the biosynthesis of naringenin-related flavonoids, a field that will be industrially explored in the next few years. The knowledge gained on the heterologous expression of type III PKSs and other modifying enzymes in yeast such as polyketide cyclases/aromatases and glycosyl transferases may be exploited in bioconversion processes; this is the case with the production of vanillin from ferulic acid [111]. Particularly important for the food industry is the debittering of the grape fruit juice by naringenase complexes [112,113]. Finally, the recent availability of naringenin biosynthesis genes in *Streptomyces clavuligerus* and other actinobacteria opens the possibility to construct highly efficient expression systems of flavonoid genes in model actinobacteria with well-developed genetic manipulation systems as platforms for their industrial exploitation.

## Figures and Tables

**Figure 1 antibiotics-11-00082-f001:**
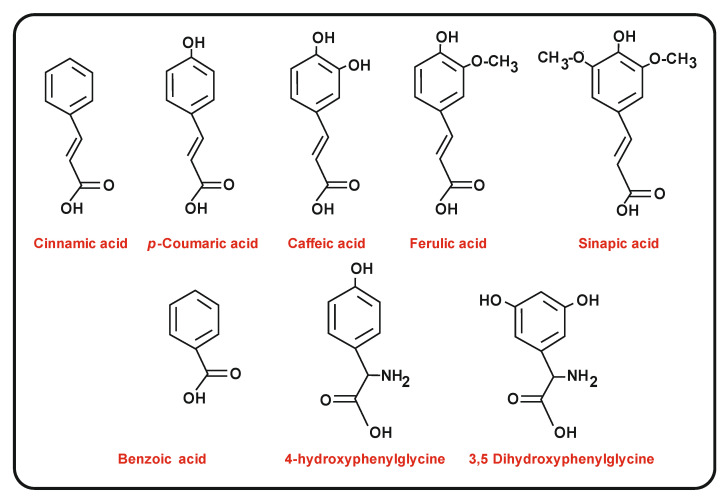
Aromatic acids and non-proteinogenic amino acids utilized by chalcone synthases (CHSs) as starter units in the phenylpropanoid pathway by different actinobacteria.

**Figure 2 antibiotics-11-00082-f002:**
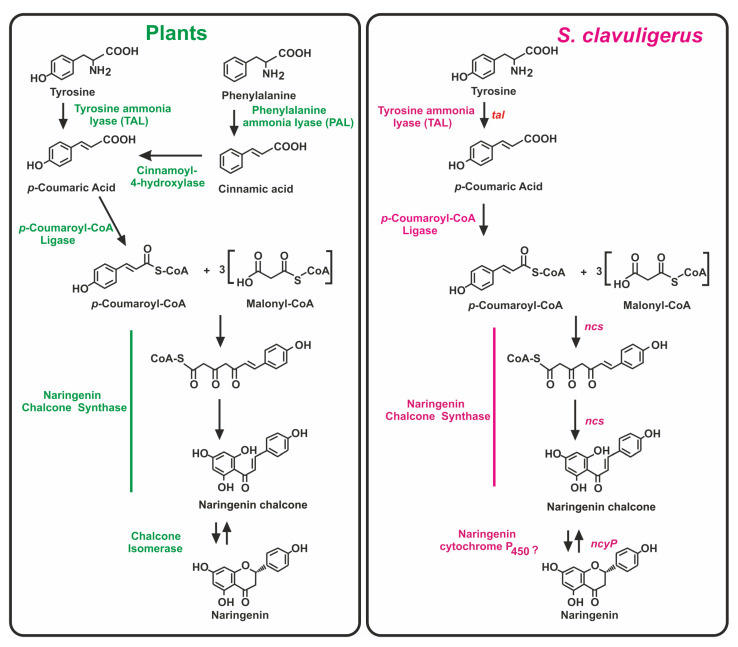
Biosynthetic pathway of naringenin in plants (green letters, left panel) and in *Streptomyces clavuligerus* (red letters, right panel). The starter units and the intermediates are indicated below the structures. The enzymes involved in every step of the pathway are labelled in colour. Note that either tyrosine or phenylalanine are used as precursors of the starter units in plants, but only tyrosine is used in *S. clavuligerus*.

**Figure 3 antibiotics-11-00082-f003:**
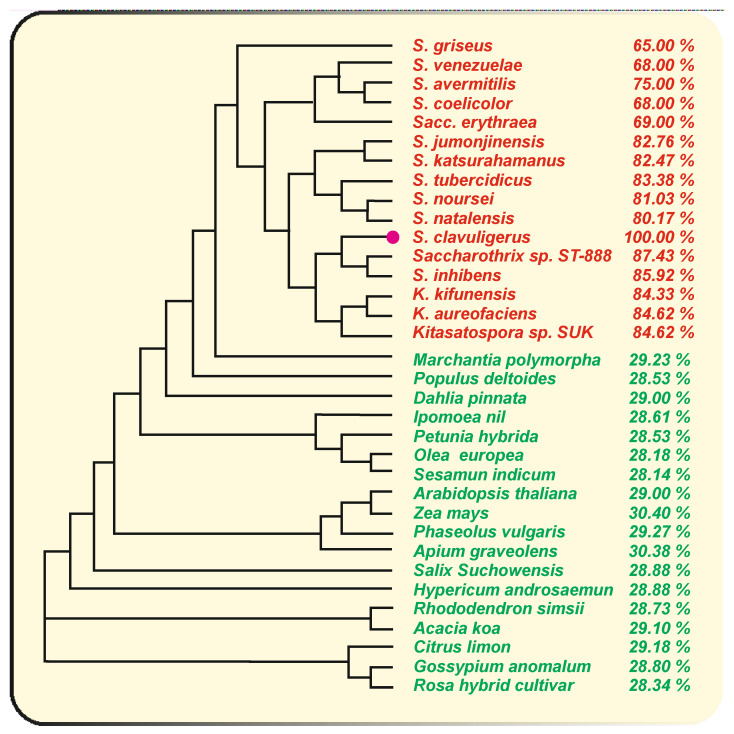
Phylogenetic three of chalcone synthases of actinobacteria (red colour) and plants (green colour). The name of the organism is indicated at the right side of its location in the phylogenetic tree and the identity percentage of each CHS to *S. clavuligerus* Ncs is indicated. The accession number for the actinobacteria chalcone synthases are: *S. griseus sub. griseus* NBRC 13,350 (BAG23449), *S. venezuelae* ATCC 10,712 (CCA58653), S. *avermitilis* MA-4680 (BAC74842), *S*. *coelicolor* A3(2) (CAC01488), *Saccharopolyspora erythraea* (AAL78053), *S. jumonjinensis* (WP_153523942), *S. katsurahamanus* (WP_153480465), *S. tubercidicus* (WP_159748865), *S. noursei* (WP_073449908), *S*. *natalensis* (WP_030063035), *S. clavuligerus* (WP_003962674), *Saccharothrix* sp. ST-888 (WP_045299660) *S. inhibens* (WP_128512316), *K. kifunensis* (WP_34749), *K. aureofaciens* (WP_033347252), *Kitasatospora* sp. SUK 42 (WP_196948006). The accession number of the plant chalcone synthases are: *Marchantia polymorpha* (PTQ40452), *Populus deltoides* (KAF9867454), *Dahlia pinnata* (BAJ14518), *Ipomoea nil* (XP_5608), *Petunia x hybrida* (P22928), *Olea europaea* (XP_022858585), *Sesamum indicum* (XP_011091402), *A**rabidopsis thaliana* (CAC80090), *Zea mays* (NP_001149022), *Phaseolus vulgaris* (XP_007161599), *Apium graveolens* (AGM46641), *Salix suchowensis* (KAG5249740), *Hypericum androsaemum* (Q9FUB7), *Rhododendron simsii* (KAF7132615), *Acacia koa* (AOX49211), *Citrus limon* (608.1), *Gossipium anomalum* (KAG8493015), *Rosa hybrid cultivar* (BAC66467). The phylogenetic three was obtained using the Clustal Omega Program (https://www.ebi.ac.uk/Tools/msa/clustalo/ (accessed on 20 November 2021)).

**Figure 4 antibiotics-11-00082-f004:**
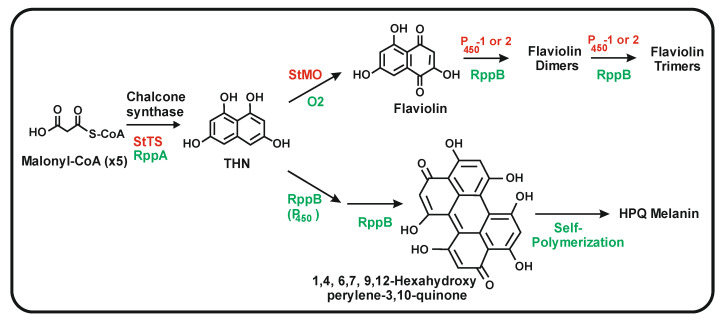
Branched biosynthetic pathway for the formation of flaviolin (above) and HPQ melanin (below). The names of the starter units and the intermediates are indicated next to their chemical structure. The enzymes involved in the pathway in *Streptomyces toxytricini* (in red letters) or in *Streptomyces coelicolor* (in green letters) are shown. Note that RppB is proposed to be involved in more than one biosynthetic step.

**Figure 5 antibiotics-11-00082-f005:**
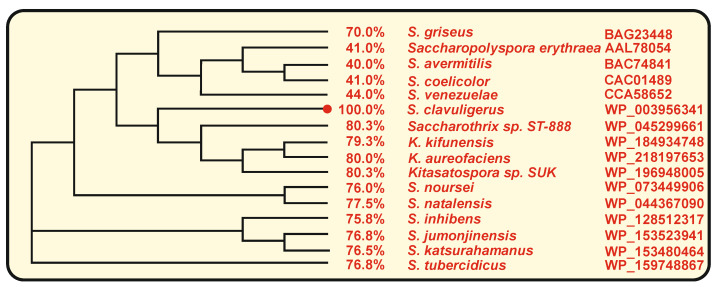
Phylogenetic tree of cytochrome P**_450_** monooxygenases. The identity percentage of each P**_450_** with respect to *S. clavuligerus* NcyP, the name of the actinobacteria and its accession number is indicated next to each branch. The phylogenetic tree was obtained using the Clustal Omega Program (https://www.ebi.ac.uk/Tools/msa/clustalo/ (accessed on 20 November 2021)).

**Figure 6 antibiotics-11-00082-f006:**
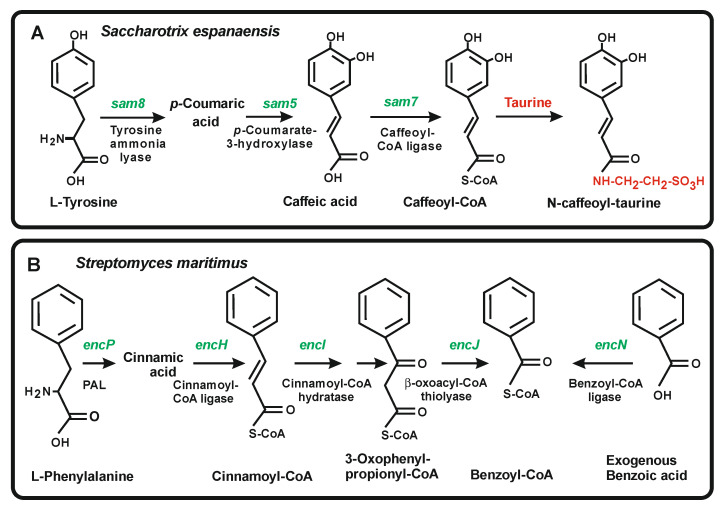
Biosynthetic steps leading from the precursors to the CoA-activated starters used by the chalcone synthases. Early steps in saccharomicins biosynthesis by *Saccharotrix espanaensis* (**panel A**) and in enterocins biosynthesis by *Streptomyces maritimus* (**panel B**). The names of the starter units, the intermediates and the enzymes involved in every step are indicated. The genes encoding the enzymes are shown in green. Note that benzoyl-CoA can be formed in *S. maritimus* either from endogenous 3-oxophenylpropionyl-CoA or from exogenous benzoic acid (see text).

**Figure 7 antibiotics-11-00082-f007:**
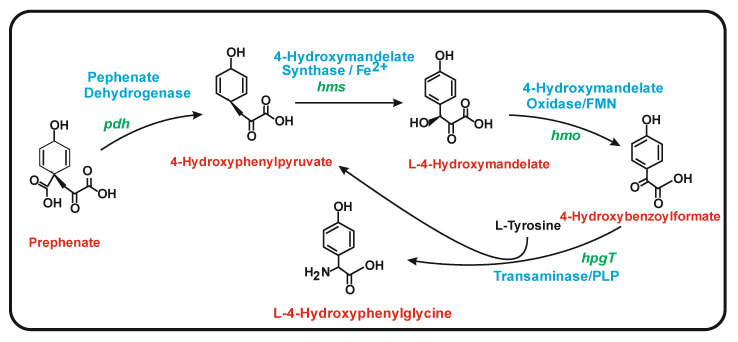
Biosynthesis of L-4-hydroxyphenylglycine. The substrate, intermediates and final product of the reaction are indicated (red colour). The enzymes involved are indicated in blue colour. The encoding genes in *Amycolatopsis orientalis* are indicated in green color. Note that the 4-hydroxyphenylpyruvate formed from the tyrosine amino donor is recycled as an intermediate of the pathway.

**Figure 8 antibiotics-11-00082-f008:**
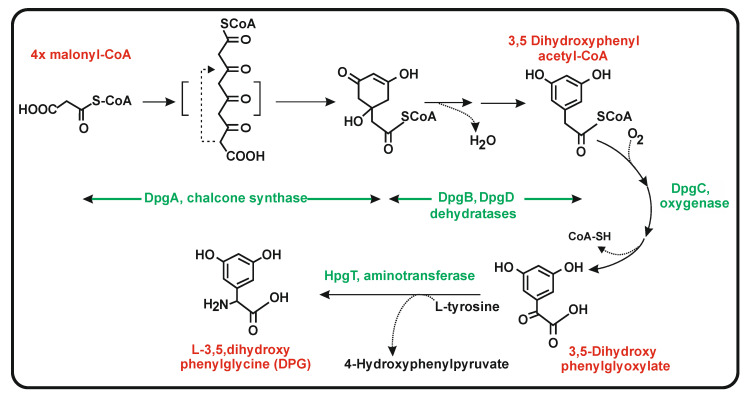
Biosynthesis of L-3,5, dihydroxyphenylglycine. The name of the enzyme involved in every step is indicated in green letters. Some steps have been proposed to consist of more than one reaction [98]. Only the putative intermediate formed by DpgA is shown.

**Figure 9 antibiotics-11-00082-f009:**
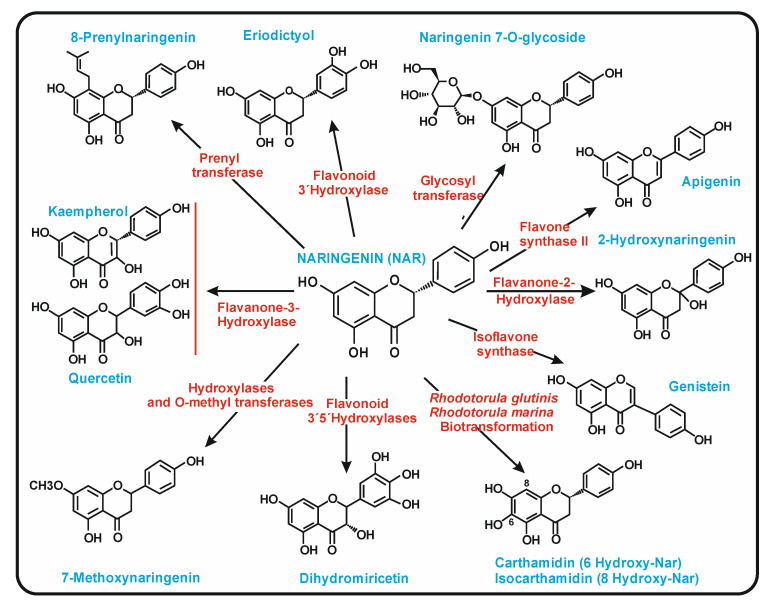
Compounds produced from naringenin using recombinant yeast strains or by biotransformation with species of *Rhodotorula.* The name of each compound is indicated in blue letters next to its chemical structure. The exogenous enzymes encoded by the recombinant yeasts, or the *Rhodotorula* species used in the biotransformation, are indicated in red letters.

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
