# Peer review of "Comparative Molecular Mechanisms of Biosynthesis of Naringenin and Related Chalcones in Actinobacteria and Plants: Relevance for the Obtention of Potent Bioactive Metabolites"

_antibiotics, 2022, doi:10.3390/antibiotics11010082_

Round 1

Reviewer 1 Report

It is a very innovative way to summarize the biosynthesis of chalcones derived from plant origin and actinomycete together, which is also more comprehensive. The manuscript summarizes the relevant content in detail and comprehensively. Nevertheless, the way the manuscript is organized is acceptable, and the internal logic of the 10 titles is confusing. In fact, the organization is not clear enough. This is the most prominent shortcoming of this manuscript. In addition, the comprehensive molecular mechanism in the title is also insufficiently reflected in the manuscript. From my point of view, the molecular mechanism is reflected in the enzyme's catalysis of the substrate, and the author should pay more attention to this.

Author Response

Reviewer 1 Comments

It is a very innovative way to summarize the biosynthesis of chalcones derived from plant origin and actinomycete together, which is also more comprehensive. The manuscript summarizes the relevant content in detail and comprehensively. Nevertheless, the way the manuscript is organized is acceptable, and the internal logic of the 10 titles is confusing. In fact, the organization is not clear enough. This is the most prominent shortcoming of this manuscript. In addition, the comprehensive molecular mechanism in the title is also insufficiently reflected in the manuscript. From my point of view, the molecular mechanism is reflected in the enzyme's catalysis of the substrate, and the author should pay more attention to this.

Answer: Thank you for your positive comments on the nature of the comparison between chalcone synthases of plants and actinobacteria. Regarding the organization of the content we have added new sentences at the end of the Introduction (lines 85-93) explaining the organization of the article that corresponds exactly to the sequence of enzymatic activities in the biosynthesis of naringenin and related chalcones. The up to date information on enzyme activities and molecular mechanisms of naringenin biosynthesis are reviewed and, in addition, the article reviews numerous modifications in chalcones of different actinobacteria that result in the biosynthesis of bioactive molecules including antibiotics and antitumoral agents.

The manuscript has been carefully revised. Many small details have been corrected (highlighted in yellow). At the suggestions of the reviewers four new figures (Numbers 3, 5, 7, 8) have been included in the modified version of the article and Fig. S1 is now included as Fig. 9. Two new references (numbers 98 and 101) have been included in this version of the article and the following references have been accordingly re-numbered.

Reviewer 2 Report

Authors Juan F. Martín and Paloma Liras review the biosynthesis pathways of naringenin in actinobacteria and plants, similarities and differences of core enzymes, as well as opportunities to discover novel bioactive metabolites.

Comments:

  1. In the Introduction & Conclusions sections (maybe Abstract), it is helpful to specify or expand how naringenin biosynthesis correlates with the biosynthesis of antimicrobial molecules, which is the focus of the section in this special issue “Challenges and Opportunities in Antibiotic Biosynthesis and Development”.
  2. Line 29: change to bioactive, instead of biosctive.
  3. Line 52: put full name of “CoA” if it is mentioned first in the text.
  4. Line 64-65 (Figure 1 legend): mentioning chalcone synthase would be helpful.
  5. Line 103 (Figure 2): it is a bit hard to read cyan/blue letters. I suggest changing to a different color to improve readership.
  6. Line 152-153: add dash symbol to the chemical names (e.g. 1,3,6,8-tetrahydroxynaphthalene).
  7. Line 169: change “alignement” to ”alignment”
  8. Line 170: revise to “species”.
  9. Line 174-179: add reference(s) for this paragraph.
  10. Line 180-187: please give values in terms of the internal cavity sizes of different CHSs, if available.
  11. Line 193 (Figure 3): Why are there 2 RppB (green letter, bottom pathway) and 2 angles?
  12. Line 194 (legend): use a lowercase letter for “flaviolin” if an uppercase letter is not necessary.
  13. Line 311: 1) change to “aminomutase”; 2) change the last “in” to “to”; 3) also revise the “a”, “b” to the correct formats.
  14. Line 359-361: please paraphrase this sentence to increase clarity.
  15. Line 519: rephrase “this information”.
  16. Line 577: revise “relevant”.
  17. Please review the manuscript, and add the missing commas in some sentences.

Author Response

Reviewer 2 Comments

Comments and Suggestions for Authors.   Authors Juan F. Martín and Paloma Liras review the biosynthesis pathways of naringenin in actinobacteria and plants, similarities and differences of core enzymes, as well as opportunities to discover novel bioactive metabolites.

Answer: Thank you for your comments.

  1. In the Introduction & Conclusions sections (maybe Abstract), it is helpful to specify or expand how naringenin biosynthesis correlates with the biosynthesis of antimicrobial molecules, which is the focus of the section in this special issue “Challenges and Opportunities in Antibiotic Biosynthesis and Development”.

Answer:  As suggested by the reviewer we have included sentences at the end of the Abstract (lines 28-32) indicating “We provide a comprehensive view of the molecular genetics and biochemistry of chalcone synthases and their impact on the development of antibacterial and antitumor compounds and other bioactive molecules”. In addition, this is reflected in lines 80-81 of the Introduction (highlighted in the text). We introduce in the  Conclusions section a sentence with examples of the impact of this research on the biosynthesis of antibiotics (lines 638-640).

Line 29: change to bioactive, instead of biosctive.  A: Changed in line 30.

  1. Line 52: put full name of “CoA” if it is mentioned first in the text.

A: Changed in line 54 to “…of coenzyme A (CoA)-activated precursor units…”.  Also, the meaning of the CoA abbreviation has been included in the Abbreviations Section.

  1. Line 64-65 (Figure 1 legend): mentioning chalcone synthase would be helpful.

A: Changed to:  “Aromatic acids and non-proteinogenic amino acids utilized by chalcone synthases as starter units”

  1.  Line 103 (Figure 2): it is a bit hard to read cyan/blue letters. I suggest changing to a different color to improve readership. A: As suggested the letters have been changed to green
  2. Line 152-153: add dash symbol to the chemical names (e.g. 1,3,6,8-tetrahydroxynaph thalene). A: Changed in lines 158, 159 and 161
  3. Line 169: change “alignement” to ”alignment”.  A: Changed in line 202
  4. Line 170: revise to “species”. A: Changed in line 203
  5. Line 174-179: add reference(s) for this paragraph. A: The paragraph 205- 212 has been modified and a new reference (ref. 41) in line 212 has been included as requested.
  6. Line 180-187: please give values in terms of the internal cavity sizes of different CHSs, if available.

 A: We have revised the articles on structure of the enzyme. Those articles describe the distances between amino acids in the crystal structure but to be best of our knowledge there are no description of the volume of the cavity. The text has been modified in lines 217-220 indicating that the limitation is in the entry channel to the internal cavity of the active site.

  1. Line 193 (Figure 3): Why are there 2 RppB (green letter, bottom pathway) and 2 angles?

A: The pathway is branched and leads to melanin and flaviolins. There are two RppB because RppB has been proposed to perform more than one biosynthetic step. The legend of Fig (now) 4 in lines 193-196 has been changed to:

Figure 4. Branched biosynthetic pathway for the formation of flaviolin (above) and HPQ melanin (below). The names of the starter units and the intermediates are indicated next to their chemical structure. The enzymes involved in the pathway in Streptomyces toxytricini (in red letters) or in Streptomyces coelicolor (in green letters) are shown. Note that RppB is proposed to be involved in more than one biosynthetic step.

  1. Line 194 (legend): use a lowercase letter for “flaviolin” if an uppercase letter is not necessary.

A: Changed in legend of Fig (now) 4 as suggested.

  1. Line 311: 1) change to “aminomutase”; 2) change the last “in” to “to”; 3) also revise the “a”, “b” to the correct formats.

A: Changed in lines 348-349, 352, 356, 360. We are sorry, the a versus a error arise when we changed from Calibri to Palatino Lynotipe type of letters as suggested by the Journal. The symbols have been corrected all along the text, see in lines 348-49, 359, 363 and elsewhere in the text.

  1. Line 359-361: please paraphrase this sentence to increase clarity.

A: The paragraph has been changed to “p-Coumaric acid is abundantly produced in plants and may be converted to other aromatic compounds by industrial processes (76)” in lines 396-398.

  1. Line 519: rephrase “this information”.

A: Changed in line 581 to: “In summary, this information supports the proposed pathway…”

  1. Line 577: revise “relevant”.

A: Changed in line 650 to “Particularly important for the food industry…”

  1. Please review the manuscript, and add the missing commas in some sentences.

Answer: The manuscript has been carefully revised. Many small details have been corrected (highlighted in yellow). At the suggestions of the reviewers four new figures (Numbers 3, 5, 7, 8) have been included in the modified version of the article and Fig. S1 was included as Fig. 9. Two new references (numbers 98 and 101) have been included in this version of the article and the following references have been accordingly renumbered.

Reviewer 3 Report

In this review manuscript, the authors compared the pathways of naringenin biosynthesis in plants and actinobacteria. Biosynthetic pathways of chalcones related to naringenin were also reviewed.

I think the theme of this manuscript fits the aim of “Antibiotics”. However, this version of manuscript contains some errors, and is not friendly for readers of the journal. Therefore, I recommend to revise the manuscript.

[Major points]

All Greek characters in this manuscript are not correctly indicated in the pdf files for reviewers (α->a, β->b).

Number of the figures are insufficient for this manuscript. Thus, this manuscript is not friendly for readers of the journal.

I recommend to add figures such as pyrogenetic tree of CHS in plants and actinobacteria, the figure indicating the relationship of RppA and NcsA in actinobacteria, pyrogenetic tree of NcyP homolog in actinobacteria, biosynthetic pathways of HPG and DPG, etc.

Especially, the pyrogenetic tree of RppA and NcsA in actinobacteria is necessary, because the authors described the relationship of CHS for naringenin synthesis and that for red-brown pigment formation in this manuscript (I agree that this relationship is one of the important topics of this manuscript).

Line 75-77

This raises the question of whether naringenin is produced by more actinobacteria and, if so, if it is synthesized by the same pathway as in plants or if there are differences that may be of biotechnological interest.

I think this sentence is complicated. Thus, I recommend to revise the sentence (more clearly).

[Minor points]

Figure 2

Size of Figure 2 is too small compared to the other figures.

Line 447

(Fig. 4A) -> (Fig.4B)

Figure 4A.

Structure of Caffeoyl-CoA is not CoA form.

[Comments]

Does self-cyclization of naringenin chalcone occur in the naringenin biosynthesis pathway of actinobacteria? 

What is the biological function of naringenin in actinobacteria?

Author Response

Reviewer 3 Comments

Comments and Suggestions for Authors

In this review manuscript, the authors compared the pathways of naringenin biosynthesis in plants and actinobacteria. Biosynthetic pathways of chalcones related to naringenin were also reviewed.

I think the theme of this manuscript fits the aim of “Antibiotics”. However, this version of manuscript contains some errors, and is not friendly for readers of the journal. Therefore, I recommend to revise the manuscript.

Answer: Thank for your positive comments, we believe that your suggestions improved very much the article. Regarding the modifications to make the manuscript more friendly to readers we have introduced and the end of the Introduction a paragraph (lines 85-93) indicating the sequence of sections, that correlate closely with biosynthetic steps of naringenin in plants and actinobacteria. We have also included 4 new figures (Fig. 3, 5, 7, 8) according to the suggestion of the reviewer and corresponds to the phylogenetic threes of the chalcone synthases in plants and actinobacteria (Fig. 3), and to the P450 monooxygenases that are associated to them (Fig. 5). Figs 7 and 8 show the biosynthetic pathway of the hydroxyphenylglycine and dihydroxyphenylglycine, respectively. In addition, the former Fig S1 has been included in the text as Fig 9 as suggested by the reviewers. Also, to make the manuscript easier for non-specialist readers we have avoided the use of unusual abbreviations; the standard abbreviations used are listed in the Abbreviation Section of the article and are indicated in the text the first time that they are used.

[Major points]

1-All Greek characters in this manuscript are not correctly indicated in the pdf files for reviewers (α->a, β->b).

Answer: We are sorry, the error arise when we changed from Calibri to Palatino Lynotipe type of letters as suggested by the Journal. The symbols have been corrected all along the text.

2-Number of the figures are insufficient for this manuscript. Thus, this manuscript is not friendly for readers of the journal. I recommend to add figures such as pyrogenetic tree of CHS in plants and actinobacteria, the figure indicating the relationship of RppA and NcsA in actinobacteria, pyrogenetic tree of NcyP homolog in actinobacteria, biosynthetic pathways of HPG and DPG, etc.

Especially, the pyrogenetic tree of RppA and NcsA in actinobacteria is necessary, because the authors described the relationship of CHS for naringenin synthesis and that for red-brown pigment formation in this manuscript (I agree that this relationship is one of the important topics of this manuscript).

Answer: As indicated above the phylogenetics threes of naringenin chalcone synthases and P450 monooxygenases have been included (new figures 3 y 5). The relationship of RppA and NcsA in actinobacteria, can be observed by comparison of Figures 2 (NcsA) and new Figure 4 (RppA). A pathway of HPG and DPG is included (Figures 7 and 8 ).

 3-Line 75-77

This raises the question of whether naringenin is produced by more actinobacteria and, if so, if it is synthesized by the same pathway as in plants or if there are differences that may be of biotechnological interest.

I think this sentence is complicated. Thus, I recommend to revise the sentence (more clearly).

Answer: The paragraph has trimmed down in lines 76-78 as follows: “This raises the question of whether naringenin is produced by more actinobacteria and whether the pathway is identical to that of plants”.

[Minor points]

Figure 2. Size of Figure 2 is too small compared to the other figures.

  1. The size of Fig. 2 has been increased.

Line 447 (Fig. 4A) -> (Fig.4B).

A: Figure 4 is now Fig. 6. In line 485 it has been changed to Fig. 6A to Fig. 6B

Figure 4A. Structure of Caffeoyl-CoA is not CoA form.

A: The figure (now Fig. 6) has been corrected.

The manuscript has been carefully revised. Many small details have been changed (highlighted in yellow). Two new references (numbers 98 and 101) have been included in this version of the article and the following references have been accordingly re-numbered.

[Comments]

Does self-cyclization of naringenin chalcone occur in the naringenin biosynthesis pathway of actinobacteria? The available information indicates that the chalcone synthase complete the cyclization step, although may require the help of the P450 NcyP; however additional biochemical information is required.

What is the biological function of naringenin in actinobacteria? The biological role of naringenin in plants is well known (see Introduction). In the actinobacteria it may also partially protect the cells against UV radiation; there is not enough evidence on this role.

Round 2

Reviewer 3 Report

I have realized that the authors have properly revised the manuscript suitable for acceptance.

Only one error was found in this manuscript.

I recommend to correct this error before publication.

Line 580

L-tyrosine (“L” should be small capital letter)